# Viral kinetics of sequential SARS-CoV-2 infections

Stephen M. Kissler [1], James A. Hay [2], Joseph R. Fauver[3], Christina Mack[4], Caroline G. Tai[4], Deverick J. Anderson[5], David D. Ho [6], Nathan D. Grubaugh[7] & Yonatan H. Grad [1]✉

The impact of a prior SARS-CoV-2 infection on the progression of subsequent infections has been unclear. Using a convenience sample of 94,812 longitudinal RT-qPCR measurements from anterior nares and oropharyngeal swabs, we identified 71 individuals with two well-sampled SARS-CoV-2 infections between March 11th, 2020, and July 28th, 2022. We compared the SARS-CoV-2 viral kinetics of first vs. second infections in this group, adjusting for viral variant, vaccination status, and age. Relative to first infections, second infections usually featured a faster clearance time. Furthermore, a person's relative (rank-order) viral clearance time, compared to others infected with the same variant, was roughly conserved across first and second infections, so that individuals who had a relatively fast clearance time in their first infection also tended to have a relatively fast clearance time in their second infection (Spearman correlation coefficient: 0.30, 95% credible interval (0.12, 0.46)). These findings provide evidence that, like vaccination, immunity from a prior SARS-CoV-2 infection shortens the duration of subsequent acute SARS-CoV-2 infections principally by reducing viral clearance time. Additionally, there appears to be an inherent element of the immune response, or some other host factor, that shapes a person's relative ability to clear SARS-CoV-2 infection that persists across sequential infections.

An estimated 65% of the US population had at least two SARS-CoV-2 infections by November 2022, but the impact of prior infection on disease course in subsequent infections has been debated[1]. Some evidence indicates SARS-CoV-2 infection provides a temporary reduction in re-infection risk[2] and a durable reduction in the risk of COVID-19-related hospitalization and death[3], while a handful of studies suggest that an initial SARS-CoV-2 infection may limit recovery from COVID-19 in later SARS-CoV-2 infections[4]. These contrasting findings may result from biases that can arise in population-level studies when differences in exposure history, vaccination status, and comorbidities

are not fully accounted for. Controlling for such factors is a major challenge given geographically and temporally heterogeneous interventions, whereas examining the dynamics of SARS-CoV-2 infections at the individual level can facilitate adjusting for these biases.

Reverse transcription quantitative PCR (RT-qPCR) conducted from clinical samples collected at multiple time points during an infection offers an objective, quantitative metric of SARS-CoV-2 kinetics and can inform on key aspects of immune response and clinical progress. Such data have been used to specify how vaccination history, antibody titer and viral lineage together shape SARS-CoV-2

¹Department of Immunology and Infectious Diseases, Harvard T.H. Chan School of Public Health, Boston, MA, USA. ²Big Data Institute, Li Ka Shing Centre for Health Information and Discovery, University of Oxford, Oxford, UK. ³Department of Epidemiology, University of Nebraska Medical Center, Omaha, NE, USA. ⁴IQVIA, Durham, NC, USA. ⁵Duke Center for Antimicrobial Stewardship and Infection Prevention, Durham, NC, USA. ⁶Vagelos College of Physicians and Surgeons, Columbia University, New York, NY, USA. ⁷Department of Epidemiology of Microbial Diseases, Yale School of Public Health, New Haven, CT, USA. ✉e-mail: ygrad@hsph.harvard.edu

proliferation and clearance during an acute infection[5], which in turn can inform the clinical management of COVID-19[6] and help interpret epidemiological trends[7]. Viral kinetics therefore offer a promising metric for clarifying the impact of an initial SARS-CoV-2 infection on subsequent infections and for translating those findings into medical and public health guidance.

The impact of vaccination and variant on SARS-CoV-2 viral kinetics have been well described elsewhere[8–12]. Infections with Delta lineages feature a higher peak viral concentration than Alpha or Omicron infections, and vaccination speeds up the clearance of SARS-CoV-2 across lineages[11]. However, the impact of SARS-CoV-2 infection-conferred immunity on peak viral concentration, viral proliferation, and viral clearance in subsequent infections is less well characterized. Furthermore, it has been unclear to what extent attributes of SARS-CoV-2 kinetics, such as peak viral load or clearance rate, persist across an individual's successive infections.

Here, we collected and analyzed 94,812 SARS-CoV-2 RT-qPCR viral concentration measurements taken from longitudinal clinical samples in players, staff, and affiliates of the National Basketball Association (NBA) between March 11, 2020, and July 28, 2022. For the subset of individuals who were infected twice during the study period ($n = 71$),

we measured changes in viral kinetics between first and second infections and determined the extent to which viral kinetic features persisted across infections.

## Results

### Summary of recorded infections

During the data collection period, 3346 infections were identified among 3021 individuals. These infections reflected the timing, intensity, and lineage composition of SARS-CoV-2 transmission in the broader United States. Of these infections, we identified 1989 "well-documented" infections that were sufficiently sampled to infer viral kinetics (Supplementary Table 1 and Supplementary Table 2), as defined by at least one RT-qPCR test with cycle threshold (Ct) value under 32 and three tests with Ct values under 40[11]. One individual had four total infections, and we omitted their third and fourth from the analysis. In total, there were 71 individuals who had two well-documented infections (Fig. 1 and Table 1). These 71 individuals were the primary focus of our analysis. We used a piecewise linear model, described previously[11], to estimate the mean viral proliferation time (time from first PCR detectability to peak viral load), clearance time

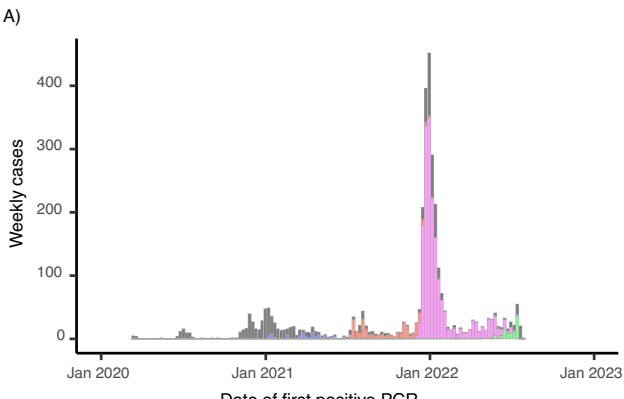

A)

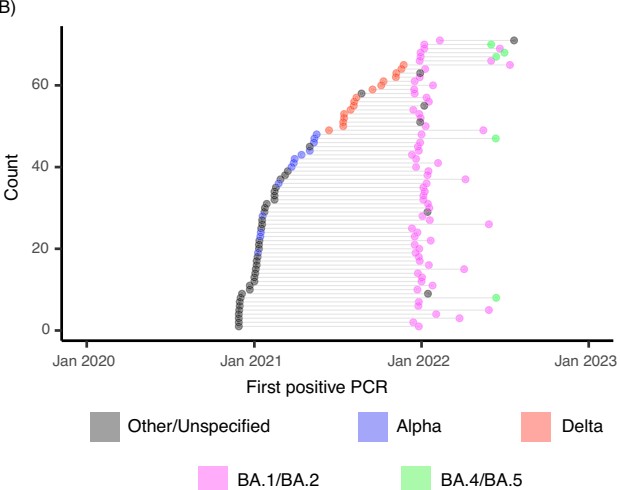

B)

**Fig. 1 | Onset times of repeat and overall infections in the dataset. A** Histogram of first positive test dates for all recorded infections in full dataset ($n = 3346$). Colors in both panels correspond to the SARS-CoV-2 variant category (Other/Unspecified: Black; Alpha: Blue; Delta: Red; BA.1/BA.2: Magenta; BA.4/BA.5: Green), where Other/Unspecified include all non-Alpha, Delta, and Omicron lineages and any samples that could not be sequenced. **B** Date of the first positive test (points) for well-documented infections in individuals with two well-documented infections ($n = 71$). Horizontal lines connect the points that correspond to infections that belong to the same person.

**Table 1 | Infection characteristics for the 71 individuals with two well-documented infections**

| Statistic | Infection number | Category | N | % |
|---|---|---|---|---|
| Total | 1 | – | 71 | 100 |
| | 2 | – | 71 | 100 |
| Variant category | 1 | Alpha | 13 | 18.3 |
| | | Delta | 16 | 22.5 |
| | | BA.1/BA.2 | 6 | 8.5 |
| | | BA.4/BA.5 | 0 | 0 |
| | | Other/unspecified | 36 | 50.7 |
| | 2 | Alpha | 0 | 0 |
| | | Delta | 0 | 0 |
| | | BA.1/BA.2 | 60 | 84.5 |
| | | BA.4/BA.5 | 5 | 7.0 |
| | | Other/unspecified | 6 | 8.5 |
| Vaccination status | 1 | Not vaccinated | 43 | 60.6 |
| | | Fully vaccinated | 18 | 25.4 |
| | | Not reported | 10 | 14.1 |
| | 2 | Not vaccinated | 1 | 1.4 |
| | | Fully vaccinated | 60 | 84.5 |
| | | Not reported | 10 | 14.1 |
| Booster status | 1 | Not boosted | 47 | 66.2 |
| | | Boosted | 6 | 8.5 |
| | | Not reported | 18 | 25.4 |
| | 2 | Not boosted | 10 | 14.1 |
| | | Boosted | 43 | 60.6 |
| | | Not reported | 18 | 25.4 |
| Age group | 1 | [0, 30) | 40 | 56.3 |
| | | [30, 50) | 28 | 39.4 |
| | | [50, 100) | 3 | 4.2 |
| | 2 | [0, 30) | 37 | 52.1 |
| | | [30, 50) | 31 | 43.7 |
| | | [50, 100) | 3 | 4.2 |

Counts are listed by variant category, vaccination status, booster status, and age group, each stratified by infection cardinality (first or second). Well-documented infections are those with at least one RT-qPCR Ct <32 and three Ct <40 (the limit of detection). Infections occurred between 11 March 2020 and 28 July 2022.

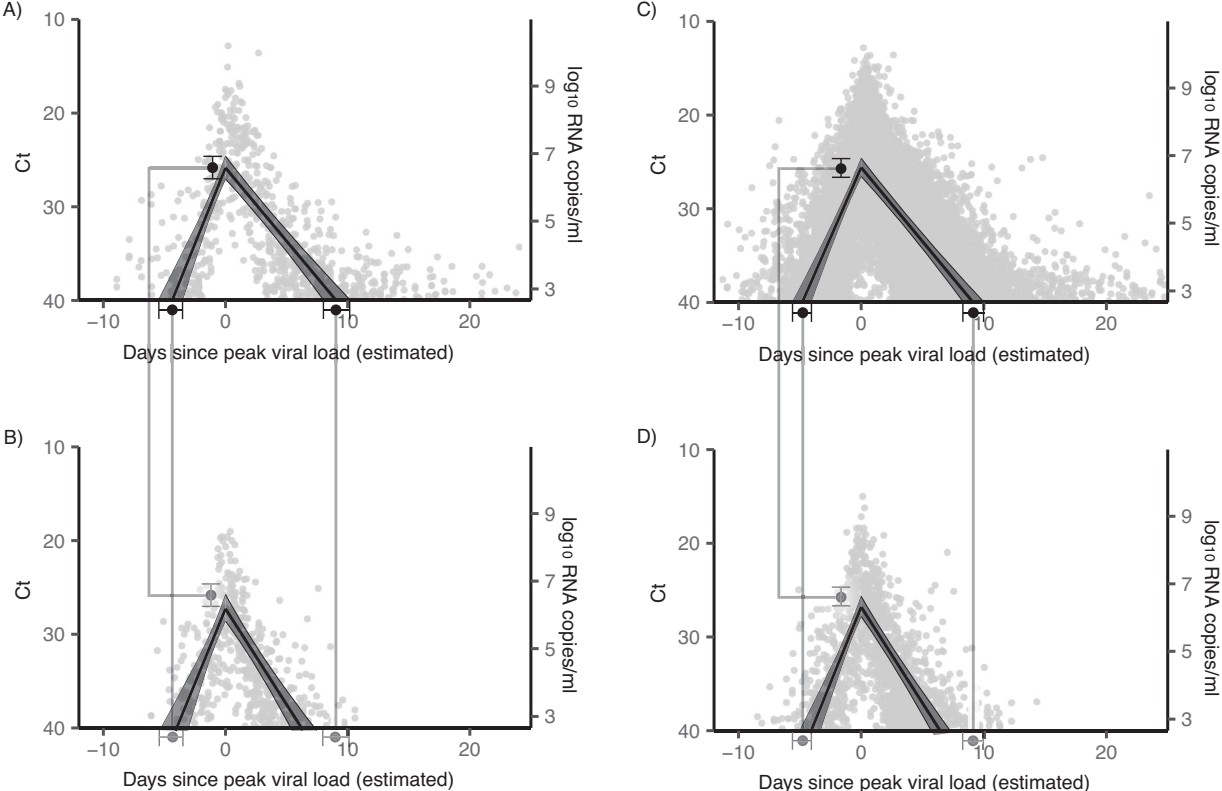

**Fig. 2 | Viral kinetics of first vs. second infections. A, B** Mean posterior viral trajectory (solid lines) with 95% credible interval (shaded region) for well-documented first infections (**A**) and second infections (**B**) in the 71 individuals with two well-documented infections. **C, D** Mean posterior viral trajectory (solid lines) with 95% credible interval (shaded region) for all well-documented first infections ($n = 1796$, **C**) and second infections ($n = 193$, **D**) in the dataset. In all panels, gray points depict the measured viral concentration for a single test. For each person, the points were shifted horizontally so that the individual's mean posterior peak viral concentration sits at day 0. Black points and whiskers (**A**, **C**) depict the mean and 95% credible interval for the proliferation time, peak viral concentration, and clearance time, from left to right, for first infections. These values are repeated in gray on the lower plots (**B**, **D**) to facilitate comparison with the viral kinetics of second infections.

(time from peak viral load to the end of PCR detectability), and peak viral load (maximum viral concentration) in the well-documented infections. We adjusted these estimates by the age and vaccination status of the infected individual and by the viral variant category (Alpha, Delta, BA.1/BA.2, BA.4/BA.5, and other/unspecified). These adjustments were informed by the full set of 1989 well-documented infections ("Methods").

**Second infections are cleared more quickly than first infections**
For the 71 individuals with two well-documented infections, the mean adjusted clearance time of the first infection was 9.2 days (95% credible interval: 8.1, 10.3) vs. 6.3 days (5.3, 7.4) for the second infection (Fig. 2A, B). There was no significant difference between the proliferation time or peak viral load between first and second infections (Supplementary Table 3).

The accelerated clearance time for second vs. first infections also held more generally. Across all first infections ($n = 1796$), the mean adjusted clearance time was 9.3 days (8.5, 10.2), while across all second infections (regardless of whether the first infection was well-documented in our dataset; $n = 193$), the mean adjusted clearance time was 6.6 days (5.8, 7.3) (Fig. 2C, D and Supplementary Table 4).

For the 71 individuals with two well-documented infections, we did not detect significant differences in viral kinetics of the second infection according to vaccination status (Supplementary Table 5). Again, this held more generally: across all well-documented second infections ($n = 1796$), we did not detect significant differences in viral kinetics according to vaccination status (Supplementary Table 6).

**No evidence that the kinetics of a second infection differ according to the first infection's lineage**
For the 71 individuals with two well-documented infections, we did not detect significant differences in the viral kinetics of the second infection based on the variant category of a first infection (Supplementary Table 7). This finding also held more generally: for all individuals with a well-documented second infection (including those with and without a well-documented first infection; $n = 193$), the clearance time was similar regardless of the variant category of the first infection (Supplementary Table 8).

**An individual's relative clearance speed is roughly preserved across infections**
For the 71 individuals with two well-documented infections, adjusted clearance times in first and second infections where correlated (Pearson correlation coefficient: 0.26 (0.09, 0.43); Spearman correlation coefficient, 0.30 (0.12, 0.46)). In contrast, we found no evidence of correlation between peak viral loads or proliferation times in first vs. second infections (Fig. 3 and Supplementary Table 9).

## Discussion
In individuals with multiple infections, second infections were cleared more quickly than first infections. Furthermore, one's relative speed of clearing infection roughly persisted across infections. Those with a relatively fast clearance speed in their first infection tended to have a relatively fast clearance speed in their second infection, and vice versa. Thus, while prior infection and vaccination can modulate a person's viral kinetics in absolute terms, there may also exist some further

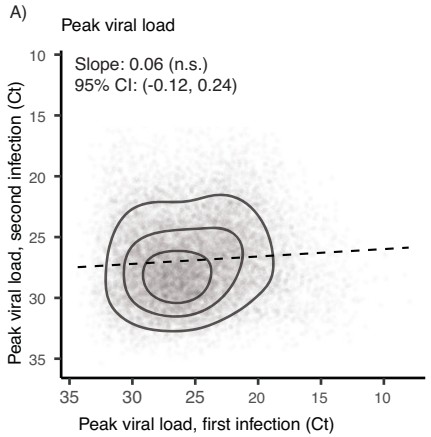

A)

Peak viral load

Slope: 0.06 (n.s.)
95% CI: (-0.12, 0.24)

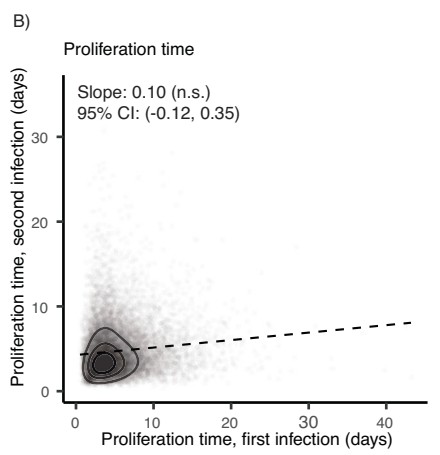

B)

Proliferation time

Slope: 0.10 (n.s.)
95% CI: (-0.12, 0.35)

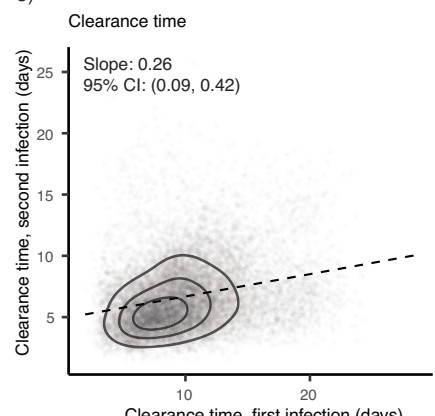

C)

Clearance time

Slope: 0.26
95% CI: (0.09, 0.42)

**Fig. 3 | Association between first- and second-infection viral kinetics.** Scatterplots of the adjusted, model-estimated **A** peak viral load, **B** proliferation time, and **C** clearance time for second infections (vertical axis) vs. first infections (horizontal axis) in the 71 individuals with two well-documented infections. Each point depicts a single posterior draw for the corresponding viral kinetic parameter belonging to a single person's first and second recorded infection, across a total of 200 draws.

Dashed lines depict the least squares linear regression to these posterior values (points), such that a positive slope indicates a positive correlation between the viral kinetic parameter between first and second infections. Posterior estimates and 95% credible intervals for the slope are listed at the top of each plot, where "n.s." denotes "not significant," corresponding to a 95% credible interval that crosses 0. Contours aid in visualizing the density of the posterior draws.

immunological mechanism, conserved across sequential infections, that determines one's strength of immune response against SARS-CoV-2 relative to others in the population.

The mechanism underlying this persistence in clearance speed rank across subsequent infections is unclear. Some possibilities include the recency of exposure to a different related coronavirus (e.g., HKU1 or OC43)[13], immune imprinting from early-lifetime exposure to certain coronavirus lineages[14], or an inherent, genome-mediated aspect of immune response. It is also unclear whether one's relative ability to clear SARS-CoV-2 infection generalizes to other coronaviruses or to other pathogens. Serological studies and genome-wide association studies may help to illuminate the mechanisms behind persistence in SARS-CoV-2 clearance time. Such studies would be valuable for improving our basic understanding of immune response to respiratory pathogens and for developing personalized clinical respiratory disease management protocols.

A consistent finding between this and other studies on SARS-CoV-2 viral kinetics is that prior antigenic exposure, through infection or vaccination, tends to speed up viral clearance, and thus to reduce the duration of test positivity[5,11,15]. The duration of viral positivity has various consequences both for clinical management and for public health surveillance. For clinical management, test results should be interpreted in the context of a patient's immune history, which can modulate both the extent and expected duration of viral shedding[5,16]. It may

also be possible to adjust the recommended duration of post-infection isolation based on infection history. When estimating epidemic growth rates using cross-sectional RT-qPCR test results, it is critical to account for immune-mediated shifts in the asymmetry between viral clearance to viral proliferation times, since this asymmetry is a key component in determining whether an epidemic is growing or shrinking[7]. We find that the difference between viral proliferation and viral clearance times decreases in second infections due to the shortened clearance time, which may reduce certainty in epidemic growth rates derived from cross-sectional RT-qPCR-based methods.

This study is limited by various factors. The cohort is predominately young, male, and healthy. While we adjusted for age, comorbidities and other underlying health factors were not measured. We were also unable to assess the relationship between measured viral concentrations and infectious virus. This study focuses primarily on individuals who were ultimately infected twice, and these individuals may differ in important immunological and behavioral ways from those who only underwent one infection during the study period. This underscores the need for further studies that capture viral and serological kinetics in tandem. Furthermore, only a small subset of individuals—71 of the over 3000 who underwent testing in this cohort—had two well-documented infections during the study period. Because of this, the statistical power of our analysis is limited, and might explain why we found no difference in the viral kinetics of second infections

based on the lineage of the first infection. Larger studies are needed to verify whether such a link exists.

In conclusion, immunity from a first SARS-CoV-2 infection affects the viral kinetics of a second SARS-CoV-2 infection principally by speeding up viral clearance and thus shortening the overall time of acute infection. The kinetics of a second BA.1/BA.2 infection are unaffected by the lineage of the first infection. Individuals who quickly cleared their first infection also generally tended to quickly clear their second infection, despite a high degree of variation in individual clearance times, pointing towards persistence of underlying immune response across multiple infections. These findings help guide the interpretation of quantitative SARS-CoV-2 tests both clinically and for surveillance and point towards persistent individual-level immune mechanisms against SARS-CoV-2 that so far remain unexplained.

## Methods

### Study design

Between March 11, 2020, and July 28, 2022, the NBA conducted regular surveillance for SARS-CoV-2 infection among players, staff, and affiliates as part of an occupational health program. This included frequent viral testing (often daily during high community COVID-19 prevalence) using a variety of platforms, but primarily via nucleic acid amplification tests, as well as clinical assessment including case diagnosis and symptom tracking. To assess viral concentration, RT-qPCR tests were conducted when possible, using anterior nares and oropharyngeal swabs collected by a trained professional and combined into a single viral transport media. Cycle threshold (Ct) values were obtained from the Roche cobas target 1 assay. Ct values were converted to genome equivalents per milliliter using a standard curve[16]. Positive controls were run on every plate and the efficacy of the primer and probe sequences used in the assay were routinely monitored for mutations that would reduce assay sensitivity. Data on participant age and vaccination status were collected where possible. Viral lineages were assigned using whole-genome sequencing, when feasible. This resulted in a longitudinal dataset of 424,401 SARS-CoV-2 tests with clinical COVID-19 history and demographic information for 3021 individuals.

Vaccination and booster status was assigned at the time of the first positive test for each infection. Full vaccination corresponded to 14 days following either the second dose of a Pfizer or Moderna vaccine or the first dose of a Johnson and Johnson/Janssen vaccine. A person was considered "boosted" 14 days after an additional Pfizer or Moderna dose following their initial vaccination course.

### Genome sequencing and lineage assignment

Whole genome sequencing of remnant diagnostic samples was performed to determine viral lineages using an overlapping amplicon-based library preparation strategy (i.e., Primal Seq). Following previously described methods, RNA was extracted from clinical samples and confirmed as SARS-CoV-2 positive[17]. Libraries were prepared in accordance with the selected sequencing platform. For samples sequenced on the Oxford Nanopore Technologies MinION platform, following amplicon generation samples were prepared for multiplex sequencing using the Ligation Sequencing Kit (SQK-LSK114) with Native Barcoding (SQK-NBD114.24). Final libraries were sequenced to a target of 100,000 reads per sample. For samples sequenced on Illumina platforms, libraries were prepared using the amplicon-based Illumina COVIDseq Test v033 with COVID-Seq ARTIC viral amplication primer set (V4, 384 samples, cat#20065135) and sequenced 2×74 on Illumina NextSeq 550 or 2×100 on the Illumina NovaSeq600 following Illumina's documentation. The resulting FASTQ files were processed and analyzed on Illumina BaseSpace Labs using the Illumina DRAGEN COVID Lineage Application[18]; versions included were 3.5.0, 3.5.1, 3.5.2, 3.5.3, and 3.5.4. The DRAGEN COVID Lineage pipeline was run with default parameters as recommended by Illumina. Lineage assignment and phylogenetics analysis were accomplished using the most recent versions of Pangolin[19] and NextClade[20], respectively. Sequences are available at BioProject under accession number PRJNA1014408.

### Estimating viral kinetic parameters

We characterized the viral kinetics of the well-documented infections by fitting a hierarchical piecewise linear model to the viral concentration measurements on a logarithmic scale (as measured by the PCR cycle threshold, or Ct), following previous methods[5]. The model captures the viral proliferation time (i.e., time from first possible detection to peak), peak viral concentration, and viral clearance time (i.e., time from peak to last possible detection) of acute SARS-CoV-2 infections. Using this approach, the viral kinetics of an infection can be described by three "hinge" points: (1) the theoretical time of first PCR positivity $t_o$, (2) the peak viral load $\delta$ (which occurs at time $t_p$), and the theoretical time of last PCR positivity $t_r$. According to this model, the expected viral load as measured by Ct units beyond the limit of detection, $E[y]$, sits at the limit of detection prior to time $t_o$, then increases linearly to $\delta$ at time $t_p$, then decreases linearly back to the limit of detection at time $t_r$ (Supplementary Figure 1). From these values, we can derive the proliferation $\omega_p$ and clearance times $\omega_r$: $\omega_p = t_p - t_o$ and $\omega_r = t_r - t_p$.

We characterized an individual $i$'s proliferation time $\omega_{p[i]}$, clearance time $\omega_{r[i]}$, and peak viral load $\delta_{[i]}$ using the following formulae:

$$\omega_{p[i]} = \mathrm{Exp}\left[\beta_p + \beta_{p[c]} + \sum_a \beta_{p[a]} + \tau_p \eta_{p[i]}\right]\omega_p^* \qquad (1)$$

$$\omega_{r[i]} = \mathrm{Exp}\left[\beta_r + \beta_{r[c]} + \sum_a \beta_{r[a]} + \tau_r \eta_{r[i]}\right]\omega_r^* \qquad (2)$$

$$\delta_{[i]} = \mathrm{Exp}\left[\beta_\delta + \beta_{\delta[c]} + \sum_a \beta_{\delta[a]} + \tau_\delta \eta_{\delta[i]}\right]\delta^* \qquad (3)$$

Re-arranging yields the following equations:

$$\log\left(\omega_{p[i]}/\omega_p^*\right) = \beta_p + \beta_{p[c]} + \sum_a \beta_{p[a]} + \tau_p \eta_{p[i]} \qquad (4)$$

$$\log\left(\omega_{r[i]}/\omega_r^*\right) = \beta_r + \beta_{r[c]} + \sum_a \beta_{r[a]} + \tau_r \eta_{r[i]} \qquad (5)$$

$$\log\left(\delta_{[i]}/\delta^*\right) = \beta_\delta + \beta_{\delta[c]} + \sum_a \beta_{\delta[a]} + \tau_\delta \eta_{\delta[i]} \qquad (6)$$

The left-hand side of these equations are the logged multiplicative factor between the individual-level parameter value (indexed with subscript [i]) and a fixed, baseline value for these parameters (marked with *); for example, a proliferation time $\omega_{p[i]}$ of 6 days relative to a baseline value $\omega_p^*$ of 3 days would yield a logged multiplicative factor of $\log(6/2) \approx 0.7$. The choice of baseline value is arbitrary and is included here to improve the robustness of the MCMC algorithm by setting the parameters on a similar scale.

The remaining coefficients ($\beta$, $\tau$, $\eta$) are estimated from the data. The summed $\beta$ coefficients constitute the population mean for the associated parameter. Thus, the unadjusted population mean proliferation time, clearance time, and peak viral load are represented by $\beta_p$, $\beta_r$, and $\delta$, respectively. These unadjusted means are adjusted according to the cardinality of infection (first or second, represented by $\beta$ values with subscript [c]) and the age group, variant category, and vaccination status of the individual (represented by $\beta$ values with subscript [a]). Together, these $\beta$ values constitute the upper level of the hierarchical model.

The individual-level effects are obtained by multiplying $\tau$, the standard deviation of the population distribution, and $\eta$, which is a standard normal random variable drawn independently for each person $i$. This follows the non-centered model parameterization for hierarchical models advocated by Gelman et al.[21].

Each $\beta$ coefficient was assigned a Normal(0,1) prior distribution. This prior was chosen because, after exponentiating and multiplying by the fixed baseline values (in Eqs. S1–S3), the middle 98% of the prior distribution corresponds to a range of roughly one-tenth to ten times the baseline value. For example, for a fixed baseline proliferation time of $\omega_p* = 3$ days, the middle 98% of the prior unadjusted population mean distribution (corresponding to Exp[$\beta_p$] $\times$ $\omega_p*$) would cover a range of roughly 0.3 days to 30 days. Thus, we considered these to be broad, minimally informative priors. The qualitative findings from the main text were unchanged when using narrower priors of Normal(0, 0.25).

Similarly, we specified a Normal(0,1) distribution, truncated to be non-negative, as the priors for the $\tau$ coefficients. With this choice, the individual-level draws could have a standard deviation up to 10 times larger than the mean, population-level distribution (i.e., the distribution of the summed $\beta$ values), following the same logic as before.

As in prior work[5], we characterized the likelihood of observing a given $\Delta Ct(t)$ using the following mixture model:

$$
\begin{aligned}
L(y_{[it]}|\delta_{[i]}, t_{p[i]}, \omega_{p[i]}\omega_{r[i]}) = {} & (1-\lambda)[f_N(x|E[y_{[it]}|\delta_{[i]}, t_{p[i]}, \omega_{p[i]}, \omega_{r[i]}], \sigma) \\
& + I_{lod}F_N(0|E[y_{[it]}|\delta_{[i]}, t_{p[i]}, \omega_{p[i]}, \omega_{r[i]}], \sigma)] \qquad (7) \\
& + \lambda f_{Exp}(x|k)
\end{aligned}
$$

The left-hand side of the equation denotes the likelihood ($L$) of observing a given viral load for person $i$ at time $t$, $y_{[it]}$, as measured by Ct deviation from the limit of detection, given the model parameters $\delta$ (peak viral load), $t_p$ (time of peak viral load), $\omega_p$ (proliferation time), and $\omega_r$ (clearance time) for individual $i$ and time $t$. Recall that $E[y_{[it]}|\delta_{[i]}, t_{p[i]}, \omega_{p[i]}, \omega_{r[i]}]$ is the expected viral load for person $i$ at time $t$ as specified by the viral kinetic model given the parameters. Here, $\sigma$ denotes the observation noise, i.e., the variation in observed vs. expected (model-derived) viral load for a person at a given time point. This noise is also estimated from the data, using a prior distribution of $\sigma \sim$ Normal(0,1), truncated to nonnegative values. This roughly covers a range of +/− 2.5 Ct for the measurement error, which falls within the range of measurement error based on repeated viral load measurements from previous studies in the same cohort[11].

The likelihood captures two distinct processes: the viral kinetic process, denoted by the bracketed term preceded by a $(1-\lambda)$; and false negatives, denoted by the term preceded by a $\lambda$. In the bracketed term representing the modeled viral kinetic process, $f_N(x|E[y], \sigma(t))$ represents the Normal PDF evaluated at $x$ with mean $E[y]$ (generated by the model equations above) and observation noise $\sigma(t)$. $F_N(0|E[y], \sigma(t))$ is the Normal CDF evaluated at 0 with the same mean and standard deviation. This represents the scenario where the true viral load goes below the limit of detection, so that the observation sits at the limit of detection. $I_{lod}$ is an indicator function that is 1 if $y = 0$ and 0 otherwise; this way, the $F_N$ term acts as a point mass concentrated at $y = 0$. Last, $f_{Exp}(x|\kappa)$ is the Exponential PDF evaluated at $x$ with rate $\kappa$. We set $\kappa = \log(10)$ so that 90% of the mass of the distribution sat below 1 Ct unit and 99% of the distribution sat below 2 Ct units, ensuring that the distribution captures values distributed at or near the limit of detection. We did not estimate values for $\lambda$ or the exponential rate because they were not of interest in this study; we simply needed to include them to account for some small probability mass that persisted near the limit of detection to allow for the possibility of false negatives.

Model parameters were fit using a Hamilton Monte Carlo algorithm implemented in R (version 4.1.2) and Stan (version 2.21.3). Four chains were run for 2000 iterations each, and the first half of each chain was discarded as burn-in, yielding 4000 total posterior draws. Convergence was assessed using a Gelman–Rubin statistic of <1.1 for all parameters and the absence of divergent transitions. Code for the full analysis is available at https://github.com/skissler/Ct_SequentialInfections.

## Statistical approach
We assessed differences in viral kinetic parameters across category subsets (e.g., for first vs. second infections) by subtracting the relevant posterior draws and measuring the posterior probability mass of these differences that sat above/below 0, depending on the scenario. When fewer than 5% of these differenced posterior draws sat above/below zero, we took this as evidence of a significant difference.

To assess relative persistence in individual-level viral kinetic attributes across infections, we measured both the Pearson (raw) and Spearman (rank-based) correlations between the adjusted first-infection and second-infection proliferation time, clearance time, and peak viral load at the individual level. To perform the adjustment, we subtracted the model-estimated adjustments for age, variant, and vaccination status, leaving only the effects from infection cardinality and individual variation. We measured the Pearson and Spearman correlation for each of the 4000 draws from the posterior distribution generated by the Hamiltonian Monte Carlo fitting approach. This yielded a mean and 95% credible interval for the Pearson and Spearman correlations between each of the first- and second-infection viral kinetic parameters.

## Study oversight
This work was approved as "research not involving human subjects" by the Yale Institutional Review Board (HIC protocol # 2000028599), as it involved de-identified samples. This work was also designated as "exempt" by the Harvard Institutional Review Board (IRB20-1407). Informed consent for virological testing and anonymized analysis of the results was obtained from all participants.

## Reporting summary
Further information on research design is available in the Nature Portfolio Reporting Summary linked to this article.

## Data availability
All data needed to reproduce the findings in this manuscript may be accessed in the following repository: https://github.com/skissler/Ct_SequentialInfections. Consensus SARS-CoV-2 genome sequences are available in GenBank (NCBI) under accession numbers OR584338–OR587821.

## Code availability
All code needed to reproduce the findings in this manuscript may be accessed in the following repository: https://github.com/skissler/Ct_SequentialInfections. The code is also available via Zenodo at https://doi.org/10.5281/zenodo.8247724.

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

## Acknowledgements

The authors acknowledge Radhika Samant and Sarah Connolly for assistance with data curation. This work was supported in part by CDC contract #200-2016-91779, a sponsored research agreement to Yale University from the National Basketball Association contract #21-003529, and the National Basketball Players Association. S.M.K. received funding from NIH T32 training grant 2 T32 AI 7535-21 A1. The findings, conclusions, and views expressed are those of the author(s) and do not necessarily represent the official position of the Centers for Disease Control and Prevention (CDC).

## Author contributions

Conceptualization: S.M.K., J.A.H., J.R.F., C.M., C.G.T., D.D.H., N.D.G., Y.H.G. Methodology: S.M.K., J.A.H., J.R.F., N.D.G., Y.H.G. Investigation: S.M.K. Visualization: S.M.K., Y.H.G. Funding acquisition: J.R.F., N.D.G., Y.H.G. Project administration: Y.H.G. Supervision: Y.H.G. Writing—original draft: S.M.K., Y.H.G. Writing—review and editing: S.M.K., J.A.H., J.R.F., C.M., C.G.T., D.J.A., D.D.H., N.D.G., Y.H.G.

## Competing interests

S.M.K. has a consulting agreement with the NBA and Moderna Therapeutics. J.A.H. declares no competing interests. J.R.F. has a consulting agreement for Tempus and receives financial support from Tempus to develop SARS-CoV-2 diagnostic tests. C.M. is an employee of IQVIA, Real World Solutions. C.G.T. is an employee of IQVIA, Real World Solutions. D.J.A. is co-owner of Infection Control Education for Major Sports. D.D.H. declares no competing interests. N.D.G. has a consulting agreement for Tempus and receives financial support from Tempus to develop SARS-CoV-2 diagnostic tests. Y.H.G. has a consulting agreement with the NBA.
