## [Peer Review File · Nature Communications]

Viral kinetics of sequential SARS-CoV-2 infectionsREVIEWER COMMENTS

Reviewer #1 (Remarks to the Author):

The authors report a well conducted study of viral clearance during SARS-CoV-2 infection, and highlight the various impacts of past vaccination and previous infection on virus clearance times. They report a clear impact of previous infection on shortening virus clearance time, and this is true even with booster vaccination. These data and results are highly interesting, the paper is well written and the statistical analysis methods appear appropriate. I have only a few points to improve clarity:

1) It is often difficult to understand all the different comparisons the authors are making. There are different immune states prior to the first infection (unvaccinated/partially vaccinated/fully vaccinated/boosted) and those who were previously infected and not previously infected, and finally, whether individuals were boosted between the two infections. This complexity is multiplied by different variants. A good example of the lack of clarity is Page 3 "In BA.1/BA.2 infections, evidence of a previous infection correlates with faster clearance": "Even in individuals who had received a booster vaccine dose, BA.1/BA.2 clearance times were faster in individuals with a history of prior infection (5.1 days (4.5, 5.7)) than in those with no such history (7.1 days (6.7, 7.6); Supplementary Table 2)."

Do the authors mean that receiving a booster (a) at any point, (b) between first and second infection or (c) only before the first infection was associated with faster clearance?

"Omicron BA.1/BA.2 clearance rates were also faster in vaccine-boosted individuals with a history of prior infection (2.6 Ct/day (2.2, 3.0)) than in those without (2.0 Ct/day (1.8, 2.1); Supplementary Table 2)." Is this the same groups being compared as above, except on clearance rates rather than clearance time? I think so, but at first I thought it was a different set of groups being compared and thus, I initially thought this was vaccine-boosted individuals versus un-boosted individuals.

A part from improve the text to make it more clear, the authors could summarise in a figure or schematic the overall conclusions, of what the clearance rate in each subset analysed is and what the "ranking" of fastest to slowest clearance is?

On a related note, I wonder if the authors have thought to ask if there is a statistical relationship between number of exposure (vax or infection) and the clearance rate? it would be interesting if for example the number of exposures to vaccination or infection was a stronger determinant of the clearance rate?

2) The statistical modelling details are not provided in main text or supplement. It is clear the authors are using previously published Bayesian hierarchical piecewise linear model fitting approach, but the details of how this was conducted should be provided in the manuscript to make it clear for the reader without unnecessary reference to other publications. The authors should provide the explicit form of the model fitted to the data, the parameters that are estimated during model fitting (including which parameters have been given a hierarchical structure), along with the priors used for each parameter. It is also unclear what was done for CT values deemed to be below the limit of detection? How were these treated in the model? Ideally, these should be censored (i.e. not excluded, or assumed to be a point value of half LOD, but assumed to fall somewhere below the LOD – this should be easily specified in the highly flexible RStan framework used by the authors – if it hasn't already been considered.

3) The methods used for the correlation between speed of clearance in the first and second infections are very confusing. It seems that lots of the methods/sup. methods are to explain this analysis and that for some reason I couldn't follow this was not just a straight Spearman correlation of the clearance rates or times. It is unclear to me why a straight forward Spearman correlation is not used on only those individuals who have two infections? The p-value is also bootstrapped, and its unclear to me why this approach was used instead of the default analytical formula for the p-value? I feel fairly confident that the approach is not influencing the conclusions

too strongly since the authors use a range of approaches, but none the less, it would be nice to see the most basic analysis presented in the main text? i.e. take all the individuals who have two infection and look at the clearance rates/times and use a Spearman correlation to see if they are significantly associated.

On a related note, in this correlation, because the quantifies being correlated are estimates from a model fitting exercise, it is not ideal to perform a standard correlation analysis on this data – since this ignores the differing degrees of certainty in each estimated value. For example, some individuals may have more data points than others, and this may introduce biases that some individuals appear to have faster or slower clearance just because of the sampling timing. Another approach would be to fit the overall model of virus kinetics for individuals who had two infections, and include a parameter in the model (as a fixed effect) for the difference between clearance rate in the first and second infection. Then the authors could ask, in the model fitting, if there is a significant increase (on average) in the clearance rate between first and second infections? i.e. If the 95% Credible intervals don't include zero it would be good evidence in support of the significant association reported with the Spearman correlation. This would then account for the uncertainty in the model fit at the same time as asking if there is an association between first and second infection clearance rates.

Altogether, this is an interesting and novel study with unique data. The work is of excellent quality and important to the field, with only fairly minor room for improvement (mostly on clarity of the methods and results).

Reviewer #2 (Remarks to the Author):

In their manuscript “viral kinetics of sequential SARS-CoV-2 infections” the authors analyse the effect of previous SARS-CoV-2 infection on the viral load characteristics of subsequent infections taking vaccination status, age and infecting variant into account. Overall the study is very well designed and gives insights on a very interesting topic. The study makes use of a very unique setting, i.e. multiple testing of a large cohort of young healthy NBA players, that prevents the generalization of the results, however this limitation is clearly acknowledged. There are some points that need clarification or improvement before the manuscript can be accepted for publications:

Figure 1 legend: The authors write “lines that extend off the plot range to the left indicate the absence of a well-documented first infection”. While I understand what the authors mean, the sentence makes no sense since logically there can't be a second infection without a first infection. I would suggest to rephrase it and say “no well-documented pre-Omicron infection” or something similar.

Supplementary table 1 provides comprehensive information on the number of participants in all subgroups and which groups are compared but is unfortunately very difficult to understand. It would tremendously improve the manuscript if the authors could come up with a simplified scheme of the subgroups they compared and add it to figure 1 as panel C.

I am having a hard time understanding supplementary table 1. Why is the number of first infections: Other/unspecified, Alpha and delta followed by BA.1/BA.2 ($n=46+14+18=78$) different from the number of persons that had a second infection with BA.1/BA.2 preceded by Alpha/delta/Other/unspecified ($n=157$)? Shouldn't that be the same? Related to my comment above, could the authors please explain that more clearly?

Line 88-90: could the authors please indicate the definitions for mean proliferation time, clearance time and peak viral load here.

Once a participant was identified as COVID positive, did he/she undergo a standardized sampling schedule, e.g. every 2 days?

I assume that all samples were measured shortly after collection of the specimens. The authors say that the CT values were converted into GE/ml by using a standard curve. Did the authors account for a potential assay drift over such a long timer period, i.e. a CT value measured in Jan 2021 may not exactly indicate the same viral load as the same CT value in Jan 2022. I.e. was the

standard curve run several time during the study period to detect a potential assay drift and make ct-values and ge comparable between PCR run 1 year apart?

Supplementary table 2 and 3: I assume the values in brackets are 95%CI. Please indicate.

The authors measured only genome equivalents or ct values but not infectious viral load. That is an important limitation and should be mentioned in the discussion.

Reviewer #3 (Remarks to the Author):

Kissler et al. analyzed a unique longitudinal dataset of samples from people working at the National Basketball Association to compare the SARS-CoV-2 viral kinetics of first and secondary infections. The authors found that (1) the clearance time is faster in second infections, compared to first infections, that second infections tend to have a lower peak viral concentration, and (2) individual variability is consistent among infections, meaning that those with faster clearance time in their first infection also experience fast clearance time in their second infections. Kissler and colleagues concluded that past infections reduce the duration of secondary infections, likely due to reduced viral clearance time.

Overall, the article reads well, and the findings are relevant to a large audience. However, I have some concerns about how the data is presented and why the authors decided to compare individuals with secondary infections with those that did not have a well documented first infections. My main comments are described below:

* "Summary of recorded infections" section.

- This section needs to be clarified, and supplementary Table 1 is very difficult to read. For instance, the caption of supp table 1 says: "(2) Alpha/ Delta/ Other/ Unspecified first infections that were followed by any BA.1/ BA.2 infection (n = 78) vs. BA.1/ BA.2 infections that were preceded by any Alpha/ Delta/ Other/ Unspecified infection (n = 157)" Please clarify the difference between these two groups.

- Lines 90-96 "We assessed whether Omicron BA.1/BA.2 viral kinetics differed according to whether the infection was the individual's first SARS-CoV-2 infection (n = 1,241 well-documented infections) or their second infection (n = 159 well-documented infections; Supplementary Table 1, group 1), stratifying by booster status and adjusting for age. Then, for the subset of individuals with a first infection by a non-Omicron lineage (verified by whole-genome sequencing or an infection onset prior to November 11th, 2021) and a second infection with BA.1 or BA.2 (n = 235 well documented infections among 177 individuals; Supplementary Table 1, group 2), we estimated" If there are 159 well-documented second infection with Omicron BA.1/BA.2, how can it be 177 individuals with a first infection by a non-Omicron lineage and a second infection with BA.1 or BA.2? The way is currently written implies that the potential maximum is 159. Please verify.

- Figure 1B: Why is the total number described in the legend (n=235) greater than 193? Does it include second infections clinically confirmed? Please clarify in the figure legend.

- It seems that 193 individuals had well documented second infections but not necessarily well documented first infections. Since only 58 individuals have both first and second well-documented infections, the main figures should be focused on these 58 infections and leave the other analyses to the supplementary material.

* "In BA.1/BA.2 infections, evidence of a previous infection correlates with faster clearance" section

- Figure 2 includes individuals with a first infection that don't have an identified second infection (or at least well documented infections). It makes the analyses presented in this figure biased since the authors might be comparing two different populations. Would the main conclusions about

faster clearance hold if the authors present the analyses of figure 2 only considering the 58 individuals with two well-documented infections? Again, I think all the analyses should be done with the data from the individuals that have well-documented first and second infections.

- The confidence intervals in figure 2A are very narrow even though values vary widely between 0 and +/-10 days. Please explain how this is possible.

- Lines 115 – 117 “The reduction in second-infection clearance time also resulted in greater symmetry between proliferation and clearance times”. What does it mean “greater symmetry” mean in this context?

* “In individuals with multiple infections, second infections were cleared faster than first infections, especially when there was an intervening vaccine dose” section.

- Lines 123 – 126: “We found no significant difference in proliferation time or peak viral concentration between first infections caused by Alpha, Delta, or other/unspecified lineages in individuals with a later Omicron BA.1/BA.2 infection (n = 78) vs. second infections caused by Omicron BA.1 or BA.2 in individuals with a previous Alpha, Delta, or other/unspecified infection (n = 157)” Similar to a point already made before, it’s unclear to me why these values are not the same. Both groups have a second infection with omicron and a first infection with Alpha, Delta, or other/unspecified lineage. Please clarify the differences between them.

REVIEWER COMMENTS

Reviewer #1 (Remarks to the Author):

The authors report a well conducted study of viral clearance during SARS-CoV-2 infection, and highlight the various impacts of past vaccination and previous infection on virus clearance times. They report a clear impact of previous infection on shortening virus clearance time, and this is true even with booster vaccination. These data and results are highly interesting, the paper is well written and the statistical analysis methods appear appropriate. I have only a few points to improve clarity:

1) It is often difficult to understand all the different comparisons the authors are making. There are different immune states prior to the first infection (unvaccinated/partially vaccinated/fully vaccinated/boosted) and those who were previously infected and not previously infected, and finally, whether individuals were boosted between the two infections. This complexity is multiplied by different variants. A good example of the lack of clarity is Page 3 “In BA.1/BA.2 infections, evidence of a previous infection correlates with faster clearance”:
“Even in individuals who had received a booster vaccine dose, BA.1/BA.2 clearance times were faster in individuals with a history of prior infection (5.1 days (4.5, 5.7)) than in those with no such history (7.1 days (6.7, 7.6); Supplementary Table 2).”

1. Based on this and similar comments from the editor and other reviewers, we have decided to restrict the main analysis to the individuals with two well-documented infections caused by any variant (n = 71). The main findings are unchanged. We hope that keeping the focus on this smaller group throughout the analysis improves the clarity of the manuscript.

Do the authors mean that receiving a booster (a) at any point, (b) between first and second infection or (c) only before the first infection was associated with faster clearance?

2. Due to the reduced power of the more focused analysis, we no longer found a significant association between an intervening vaccine dose and faster clearance time, so we have removed this conclusion.

“Omicron BA.1/BA.2 clearance rates were also faster in vaccine-boosted individuals with a history of prior infection (2.6 Ct/day (2.2, 3.0)) than in those without (2.0 Ct/day (1.8, 2.1); Supplementary Table 2).” Is this the same groups being compared as above, except on clearance rates rather than clearance time? I think so, but at first I thought it was a different set of groups being compared and thus, I initially thought this was vaccine-boosted individuals versus un-boosted individuals.

3. Please see Point 1 on the more focused analysis.

A part from improve the text to make it more clear, the authors could summarise in a figure or schematic the overall conclusions, of what the clearance rate in each subset analysed is and what the "ranking" of fastest to slowest clearance is?

4. Please see Point 1 on the more focused analysis.

On a related note, I wonder if the authors have thought to ask if there is a statistical relationship between number of exposure (vax or infection) and the clearance rate? it would be interesting if

for example the number of exposures to vaccination or infection was a stronger determinant of the clearance rate?

5. In the revised analysis, we have also conducted more thorough adjustments for various exposure/vaccination groups. We find that, for second infections, vaccination status generates no discernible impact on viral kinetics beyond infection cardinality, though the sample size may make it difficult to detect such an effect.

2) The statistical modelling details are not provided in main text or supplement. It is clear the authors are using previously published Bayesian hierarchical piecewise linear model fitting approach, but the details of how this was conducted should be provided in the manuscript to make it clear for the reader without unnecessary reference to other publications. The authors should provide the explicit form of the model fitted to the data, the parameters that are estimated during model fitting (including which parameters have been given a hierarchical structure), along with the priors used for each parameter. It is also unclear what was done for CT values deemed to be below the limit of detection? How were these treated in the model? Ideally, these should be censored (i.e. not excluded, or assumed to be a point value of half LOD, but assumed to fall somewhere below the LOD – this should be easily specified in the highly flexible RStan framework used by the authors – if it hasn't already been considered.

6. We have now provided more details on the statistical modeling approach in the main text to address these points. Points below the LOD were censored (not excluded), such that when the modeled viral load passed below the limit of detection, the probability mass was mapped back to the limit of detection.

3) The methods used for the correlation between speed of clearance in the first and second infections are very confusing. It seems that lots of the methods/sup. methods are to explain this analysis and that for some reason I couldn't follow this was not just a straight Spearman correlation of the clearance rates or times. It is unclear to me why a straight forward Spearman correlation is not used on only those individuals who have two infections? The p-value is also bootstrapped, and it's unclear to me why this approach was used instead of the default analytical formula for the p-value? I feel fairly confident that the approach is not influencing the conclusions too strongly since the authors use a range of approaches, but none the less, it would be nice to see the most basic analysis presented in the main text? i.e. take all the individuals who have two infection and look at the clearance rates/times and use a Spearman correlation to see if they are significantly associated.

7. We agree that this was unnecessarily complicated; in the end, it was just a straightforward Spearman correlation. The more thorough adjustment structure in the model also made it easier to conduct these regressions in a straightforward way, rather than doing post-hoc stratification on variant categories. We have adjusted this part of the analysis and the associated explanations. We also use more standard methods to assess significance (conducting the regressions on each of the 4000 posterior draws and reporting credible intervals).

On a related note, in this correlation, because the quantities being correlated are estimates from a model fitting exercise, it is not ideal to perform a standard correlation analysis on this data – since this ignores the differing degrees of certainty in each estimated value. For example, some individuals may have more data points than others, and this may introduce biases that some individuals appear to have faster or slower clearance just because of the sampling timing. Another approach would be to fit the overall model of virus kinetics for individuals who had two infections, and include a parameter in the model (as a fixed effect) for the difference between

clearance rate in the first and second infection. Then the authors could ask, in the model fitting, if there is a significant increase (on average) in the clearance rate between first and second infections? i.e. If the 95% Credible intervals don't include zero it would be good evidence in support of the significant association reported with the Spearman correlation. This would then account for the uncertainty in the model fit at the same time as asking if there is an association between first and second infection clearance rates.

8. In the updated version of this analysis, since we now rely explicitly on posterior draws, this should now incorporate uncertainty based on poorly sampled trajectories.

Altogether, this is an interesting and novel study with unique data. The work is of excellent quality and important to the field, with only fairly minor room for improvement (mostly on clarity of the methods and results).

Thank you for this review!

Reviewer #2 (Remarks to the Author):

In their manuscript “viral kinetics of sequential SARS-CoV-2 infections” the authors analyse the effect of previous SARS-CoV-2 infection on the viral load characteristics of subsequent infections taking vaccination status, age and infecting variant into account. Overall the study is very well designed and gives insights on a very interesting topic. The study makes use of a very unique setting, i.e. multiple testing of a large cohort of young healthy NBA players, that prevents the generalization of the results, however this limitation is clearly acknowledged. There are some points that need clarification or improvement before the manuscript can be accepted for publications:

Figure 1 legend: The authors write “lines that extend off the plot range to the left indicate the absence of a well-documented first infection”. While I understand what the authors mean, the sentence makes no sense since logically there can't be a second infection without a first infection. I would suggest to rephrase it and say “no well-documented pre-Omicron infection” or something similar.

We have now restricted the main analysis to the 71 individuals with two well-documented infections (see Point 1 in the response to Reviewer 1), which makes the need for the extended lines unnecessary.

Supplementary table 1 provides comprehensive information on the number of participants in all subgroups and which groups are compared but is unfortunately very difficult to understand. It would tremendously improve the manuscript if the authors could come up with a simplified scheme of the subgroups they compared and add it to figure 1 as panel C.

We have removed Supplementary Table 1 in light of Point 1.

I am having a hard time understanding supplementary table 1. Why is the number of first infections: Other/unspecified, Alpha and delta followed by BA.1/BA.2 ($n=46+14+18=78$) different from the number of persons that had a second infection with BA.1/BA.2 preceded by Alpha/delta/Other/unspecified ($n=157$)? Shouldn't that be the same? Related to my comment above, could the authors please explain that more clearly?

See Point 1 in response to Reviewer 1.

Line 88-90: could the authors please indicate the definitions for mean proliferation time, clearance time and peak viral load here.

We have added these definitions.

Once a participant was identified as COVID positive, did he/she undergo a standardized sampling schedule, e.g. every 2 days?

Yes, once a participant was identified as COVID positive, most underwent continued sampling at the same cadence as prior to their detection (daily where possible).

I assume that all samples were measured shortly after collection of the specimens. The authors say that the CT values were converted into GE/ml by using a standard curve. Did the authors account for a potential assay drift over such a long timer period, i.e. a CT value measured in Jan 2021 may not exactly indicate the same viral load as the same CT value in Jan 2022. I.e. was

the standard curve run several time during the study period to detect a potential assay drift and make ct-values and ge comparable between PCR run 1 year apart?

Yes, all samples were measured as soon as possible after sample collection. The standard curve calibration was run just once, but positive controls and monitoring for mutations that could have reduced the sensitivity of the assay were continually assessed. We have added the following text to the Materials and Methods:

“Positive controls were run on every plate and the efficacy of the primer and probe sequences used in the assay were routinely monitored for mutations that would reduce assay sensitivity.”

Supplementary table 2 and 3: I assume the values in brackets are 95%CI. Please indicate. The authors measured only genome equivalents or ct values but not infectious viral load. That is an important limitation and should be mentioned in the discussion.

This interpretation is correct; we have clarified this in the Supplementary Tables and added a point in the Discussion about the lack of assessment of infectious virus.

Reviewer #3 (Remarks to the Author):

Kissler et al. analyzed a unique longitudinal dataset of samples from people working at the National Basketball Association to compare the SARS-CoV-2 viral kinetics of first and secondary infections. The authors found that (1) the clearance time is faster in second infections, compared to first infections, that second infections tend to have a lower peak viral concentration, and (2) individual variability is consistent among infections, meaning that those with faster clearance time in their first infection also experience fast clearance time in their second infections. Kissler and colleagues concluded that past infections reduce the duration of secondary infections, likely due to reduced viral clearance time.

Overall, the article reads well, and the findings are relevant to a large audience. However, I have some concerns about how the data is presented and why the authors decided to compare individuals with secondary infections with those that did not have a well documented first infections. My main comments are described below:

* “Summary of recorded infections” section.

- This section needs to be clarified, and supplementary Table 1 is very difficult to read. For instance, the caption of supp table 1 says: “(2) Alpha/ Delta/ Other/ Unspecified first infections that were followed by any BA.1/ BA.2 infection (n = 78) vs. BA.1/ BA.2 infections that were preceded by any Alpha/ Delta/ Other/ Unspecified infection (n = 157)” Please clarify the difference between these two groups.

Please see Point 1 in response to Reviewer 1. We have taken Reviewer 3's suggestion to focus the analysis on the individuals with two well-documented infections.

- Lines 90-96 “We assessed whether Omicron BA.1/BA.2 viral kinetics differed according to whether the infection was the individual's first SARS-CoV-2 infection (n = 1,241 well-documented infections) or their second infection (n = 159 well-documented infections; Supplementary Table 1, group 1), stratifying by booster status and adjusting for age. Then, for the subset of individuals with a first infection by a non-Omicron lineage (verified by whole-genome sequencing or an infection onset prior to November 11th, 2021) and a second infection with BA.1 or BA.2 (n = 235 well documented infections among 177 individuals; Supplementary Table 1, group 2), we estimated ...” If there are 159 well-documented second infection with Omicron BA.1/BA.2, how can it be 177 individuals with a first infection by a non-Omicron lineage and a second infection with BA.1 or BA.2? The way is currently written implies that the potential maximum is 159. Please verify.

Please see Point 1 in response to Reviewer 1.

- Figure 1B: Why is the total number described in the legend (n=235) greater than 193? Does it include second infections clinically confirmed? Please clarify in the figure legend.

Please see Point 1 in response to Reviewer 1.

- It seems that 193 individuals had well documented second infections but not necessarily well documented first infections. Since only 58 individuals have both first and second well-documented infections, the main figures should be focused on these 58 infections and leave the other analyses to the supplementary material.

We have taken this suggestion and used it to substantially revise the methods and results; the main results have remained the same, and we believe that this central focus on a single group of individuals has helped to clarify the manuscript.

* “In BA.1/BA.2 infections, evidence of a previous infection correlates with faster clearance” section

- Figure 2 includes individuals with a first infection that don't have an identified second infection (or at least well documented infections). It makes the analyses presented in this figure biased since the authors might be comparing two different populations. Would the main conclusions about faster clearance hold if the authors present the analyses of figure 2 only considering the 58 individuals with two well-documented infections? Again, I think all the analyses should be done with the data from the individuals that have well-documented first and second infections.

Please see Point 1 in response to Reviewer 1.

- The confidence intervals in figure 2A are very narrow even though values vary widely between 0 and +/-10 days. Please explain how this is possible.

The credible intervals here are credible intervals for the mean, which are very narrow since there is so much data informing the mean. This differs from a prediction interval, which would be much wider, as it would contain the bound within which 95% of individual trajectories would be expected to fall.

- Lines 115 – 117 “The reduction in second-infection clearance time also resulted in greater symmetry between proliferation and clearance times”. What does it mean “greater symmetry” mean in this context?

Here, “greater symmetry” means that the ratio of clearance to proliferation time was closer to 1. With little underlying immunity, most viral kinetic trajectories had a substantially longer clearance time than proliferation time, leading to a “skewed” viral trajectory. As immunity accrues, the clearance time tends to shorten more than the proliferation time does, so that the proliferation and clearance times become more similar. This is important because some methods that use viral load measurements to infer epidemic growth and decay rely on a systematic difference between proliferation and clearance times. That said, we have removed the reference to symmetry in the main text and only mention this fact in the Discussion, where it can be placed in its proper context.

* “In individuals with multiple infections, second infections were cleared faster than first infections, especially when there was an intervening vaccine dose” section.

- Lines 123 – 126: “We found no significant difference in proliferation time or peak viral concentration between first infections caused by Alpha, Delta, or other/unspecified lineages in individuals with a later Omicron BA.1/BA.2 infection (n = 78) vs. second infections caused by Omicron BA.1 or BA.2 in individuals with a previous Alpha, Delta, or other/unspecified infection (n = 157)” Similar to a point already made before, it's unclear to me why these values are not the same. Both groups have a second infection with omicron and a first infection with Alpha, Delta, or other/unspecified lineage. Please clarify the differences between them.

Please see Point 1 in response to Reviewer 1.

REVIEWER COMMENTS

Reviewer #1 (Remarks to the Author):

The authors have now added the statistical methods to the manuscript. Unfortunately, this has raised, for this reviewer, some new concerns regarding the modelling – which may impinge on the conclusions and thus should be tested/corrected:

1. My biggest concern, as it may impact the conclusions, is that the prior distributions used to fit the key model parameters - which test whether there is a difference between any groups - seem overly informative and narrow. I.e. these only effectively tolerate up to 2 fold difference from the central estimate but favouring far less than this I.e. lines 238 and 256. These narrow prior assumptions could force the conclusions of no difference where there might actually be some evidence of a difference. The authors must stress test their conclusions by varying their choice of prior. E.g. choose a much larger SD for the normal distribution (e.g. x10 larger 2.5 instead of 0.25)?

2. The authors describe the model as hierarchical. But the description doesn't support this. In lines 258 it appears the individual level parameters are independently estimated (each with the same prior), but have no higher level structure imposed on them (i.e. the parameters are not drawn from a single distribution). This is more like having individual parameters for each subject which could notionally skew the conclusions. See Bayesian Data Analysis by Gelman et al Chapter 5.

If my understanding of their model is correct, the authors should either change the text and not refer to this model as a hierarchical model (simple solution but not as ideal). Or should amend the model to include a hierarchical structure on the individual level parameters (preferred solution),

E.g. σ_i should be drawn from a distribution $N(0,S)$, where S is a fitted parameter with its own prior.

A similar note that the first two levels of this hierarchical model are not hierarchical levels. These are just covariates for each factor being explored.

3. Are the parameters ω_p^* , ω_r^* and δ^* fitted? It seems from line 237 they are not? Which would be unusual. If these are not fitted parameters - how are they derived? Why not just fit the central estimate for day 1 and the difference between day 1 and 2 as the two model parameters (a more typical parameterisation)?

Minor comments:

1. For each of the stratified analysis in Table S3, S4, S5 – the authors should include the numbers of individuals in each sub category.

E.g. "Other/None (n=XX)", "Alpha (n=XX)", etc.

2. I like the authors approach of focusing on the 71 with well documented infections and then confirming the results in the bigger data set. But I also think it's worth the authors adding additional supp. tables with these additional data (just as they have done in Table S3), e.g. can the authors make a version of table S4 and S5 that includes all infections but with the stratification by vaccine and variant respectively? (I don't expect more attention in the main text on these tables, just presentation of the results in the supplement to support some of the conclusions, e.g. line 114 says the authors conducted this analysis but doesn't give any of the tables to support the statement).

3. It would be better for the reader to understand the numbers if Table S1 was split into two, i.e. One table for well documented first infections (n=1796), and one for well documented infections second infections (n=193).

4. The authors need to caveat this conclusion more up front in the discussion – since it is very much limited by the data to detect a difference... "The lineage of the first infection did not meaningfully affect the viral kinetics of a later BA.1/BA.2 infection."

Reviewer #2 (Remarks to the Author):

The authors answered to all my comments sufficiently.

There is only one minor typo:

Line 98: I assume it should read "Figure 2A-B

Reviewer #3 (Remarks to the Author):

The authors have addressed all my comments from the first revision, and I acknowledge the manuscript is much more clear now. I have only one minor comment, which is about how the main figures are cited in the text:

* Lines 96-98: "For the 71 individuals with two well-documented infections, the mean adjusted clearance time of the first infection was 9.1 days (95% credible interval: 8.1, 10.2) vs. 6.3 days (5.4, 7.3) for the second infection (Figure 1A-B)." It should say Figure 2A-B, not Figure 1A-B.

* Lines 101-104: "The accelerated clearance time for second vs. first infections also held more generally. Across all first infections ($n = 1,796$), the mean adjusted clearance time was 9.2 days (8.4, 10), while across all second infections (regardless of whether the first infection was well-documented in our dataset; $n = 193$), the mean adjusted clearance time was 6.5 days (5.7, 7.3) (Supplementary Table 3)." If I understood correctly, the authors could also cite Figure 2C-D here (which hasn't been cited in the current version of the manuscript).

Reviewer #1 (Remarks to the Author):

The authors have now added the statistical methods to the manuscript. Unfortunately, this has raised, for this reviewer, some new concerns regarding the modelling – which may impinge on the conclusions and thus should be tested/corrected:

1. My biggest concern, as it may impact the conclusions, is that the prior distributions used to fit the key model parameters - which test whether there is a difference between any groups - seem overly informative and narrow. I.e. these only effectively tolerate up to 2 fold difference from the central estimate but favouring far less than this I.e. lines 238 and 256. These narrow prior assumptions could force the conclusions of no difference where there might actually be some evidence of a difference. The authors must stress test their conclusions by varying their choice of prior. E.g. choose a much large SD for the normal distribution (e.g. x10 larger 2.5 instead of 0.25)?

Thank you for raising these points. The prior distributions we chose were based on previous work, where sensitivity analyses of this sort were carried out, but we agree that it is worthwhile to include similar stress tests here. We have widened the prior distributions such that they now tolerate a 10-fold difference, rather than a 2-fold difference, from the central estimate. In some cases, the widths of the credible intervals increased, but the overall conclusions of the paper remain unchanged. This is now documented in the revised Materials and Methods, and we now report the means and confidence intervals obtained from these wider priors in the main text.

2. The authors describe the model as hierarchical. But the description doesn't support this. In lines 258 it appears the individual level parameters are independently estimated (each with the same prior), but have no higher level structure imposed on them (i.e. the parameters are not drawn from a single distribution). This is more like having individual parameters for each subject which could notionally skew the conclusions. See Bayesian Data Analysis by Gelman et al Chapter 5.

If my understanding of their model is correct, the authors should either change the text and not refer to this model as a hierarchical model (simple solution but not as ideal). Or should amend the model to include a hierarchical structure on the individual level parameters (preferred solution), e.g. σ_i should be drawn from a distribution $N(0, S)$, where S is a fitted parameter with its own prior.

We thank the reviewer for highlighting this issue. The model is indeed hierarchical. We have rewritten the model description to align more closely with the hierarchical modeling approach from Gelman *et al.*, which was the basis from which we originally designed our model. We now begin the model description with the individual level and work upward, rather than from the upper level of the hierarchy downward. We also have added a citation of Gelman *et al.* at the appropriate part of the model description. The model itself is structured similarly to the hierarchical "schools" example from Gelman *et al.*, Appendix C2 (pages 592-594).

A similar note that the first two levels of this hierarchical model are not hierarchical levels. These are just covariates for each factor being explored.

This is accurate; our original description was incorrect. We have edited the model description to refer now to just two levels of hierarchy.

3. Are the parameters ω_p^* , ω_r^* and δ^* fitted? It seems from line 237 they are not? Which would be unusual. If these are not fitted parameters - how are they derived? Why not just fit the central estimate for day 1 and the difference between day 1 and 2 as the two model parameters (a more typical parameterisation)?

Yes, these parameters are fit. Please see the above points.

Minor comments:

1. For each of the stratified analysis in Table S3, S4, S5 – the authors should include the numbers of individuals in each sub category.

E.g. “Other/None (n=XX)”, “Alpha (n=XX)”, etc.

Thank you; we have included these.

2. I like the authors approach of focusing on the 71 with well documented infections and then confirming the results in the bigger data set. But I also think it's worth the authors adding additional supp. tables with these additional data (just as they have done in Table S3), e.g. can the authors make a version of table S4 and S5 that includes all infections but with the stratification by vaccine and variant respectively? (I don't expect more attention in the main text on these tables, just presentation of the results in the supplement to support some of the conclusions, e.g. line 114 says the authors conducted this analysis but doesn't give any of the tables to support the statement).

We have added these tables; these are the new Supplementary Tables 6 and 8.

3. It would be better for the reader to understand the numbers if Table S1 was split into two, i.e. One table for well documented first infections (n=1796), and one for well documented infections second infections (n=193).

We have made this edit; these are now Supplementary Tables 1 and 2.

4. The authors need to caveat this conclusion more up front in the discussion – since it is very much limited by the data to detect a difference... “The lineage of the first infection did not meaningfully affect the viral kinetics of a later BA.1/BA.2 infection.”

We agree, and have moved this point and a fuller discussion of it to the “limitations” section of the Discussion.

Reviewer #2 (Remarks to the Author):

The authors answered to all my comments sufficiently.

There is only one minor typo:

Line 98: I assume it should read "Figure 2A-B

Yes, this is correct; we have made this edit.

Reviewer #3 (Remarks to the Author):

The authors have addressed all my comments from the first revision, and I acknowledge the manuscript is much more clear now. I have only one minor comment, which is about how the main figures are cited in the text:

* Lines 96-98: "For the 71 individuals with two well-documented infections, the mean adjusted clearance time of the first infection was 9.1 days (95% credible interval: 8.1, 10.2) vs. 6.3 days (5.4, 7.3) for the second infection (Figure 1A-B)." It should say Figure 2A-B, not Figure 1A-B. Thank you for pointing this out; we have made this edit.

* Lines 101-104: "The accelerated clearance time for second vs. first infections also held more generally. Across all first infections ($n = 1,796$), the mean adjusted clearance time was 9.2 days (8.4, 10), while across all second infections (regardless of whether the first infection was well-documented in our dataset; $n = 193$), the mean adjusted clearance time was 6.5 days (5.7, 7.3) (Supplementary Table 3)." If I understood correctly, the authors could also cite Figure 2C-D here (which hasn't been cited in the current version of the manuscript). This is correct, thank you; we have made this edit.

REVIEWERS' COMMENTS

Reviewer #1 (Remarks to the Author):

The authors have addressed all of this reviewers concerns in the latest revisions and the modelling methods are now much more explicit and clearly convey a hierarchical model structure. They have also now reported their result with broad priors for a difference between groups, and found the same conclusions.

This work is an important contribution to the field.